# SeViCES: Unifying Semantic-Visual Evidence Consensus for Long Video Understanding

## Abstract

Long video understanding remains challenging due to its complex, diverse, and temporally scattered content. Although video large language models (Video-LLMs) can process videos lasting tens of minutes, applying them to truly long sequences is computationally prohibitive and often leads to unfocused or inconsistent reasoning. A promising solution is to select only the most informative frames, yet existing approaches typically ignore temporal dependencies or rely on unimodal evidence, limiting their ability to provide complete and query-relevant context. We propose a **Se**mantic–**Vi**sual **C**onsensus **E**vidence **S**election (SeViCES) framework for effective and reliable long video understanding. SeViCES is training-free and model-agnostic, and introduces two key components. The Semantic–Visual Consensus Frame Selection (SVCFS) module selects frames through (1) a temporal-aware semantic branch that leverages LLM reasoning over captions, and (2) a cluster-guided visual branch that aligns embeddings with semantic scores via mutual information. The Answer Consensus Refinement (ACR) module further resolves inconsistencies between semantic- and visual-based predictions by fusing evidence and constraining the answer space. Extensive experiments on long video understanding benchmarks show that SeViCES consistently outperforms state-of-the-art methods in both accuracy and robustness, demonstrating the importance of consensus-driven evidence selection for Video-LLMs.

## 1 Introduction

Understanding long videos remains inherently challenging due to their complex, diverse, and temporally scattered content (Zou et al., 2024). Unlike short clips, long videos often span multiple events, scenes, and semantic contexts, making it difficult to capture a coherent global narrative (Wang et al., 2025a). Although recent video large language models (Video-LLMs) such as LLaVA-Video (Zhang et al., 2025d) and Qwen2.5-VL (Bai et al., 2025) demonstrate the ability to process videos lasting tens of minutes, their performance on truly long video understanding remains unsatisfactory. Directly feeding massive numbers of visual frames into Video-LLMs not only incurs prohibitive computational costs but also dilutes attention, often leading to unfocused reasoning and logically inconsistent interpretations (Zhang et al., 2025f).

A natural way to mitigate these limitations is to reduce the visual content input to Video-LLMs, allowing them to focus on the information most relevant to a query without relying on larger architectures or extensive fine-tuning. This raises a central challenge: *how to select the input visual content effectively*. The selected content must be both query-relevant and evidence-complete, filtering out redundant information while preserving the full reasoning chain required to answer the query. Existing solutions fall into two main categories: *token selection* and *frame selection*. Token selection methods (Zhang et al., 2025c;e; Shen et al., 2025a) reduce the number of visual tokens by estimating their importance or training a selector, but they typically require expensive fine-tuning or rely on layer-specific importance scores, which are not always consistent across layers. In contrast, frame selection methods (Liu et al., 2025a; Santos et al., 2025) choose a subset of frames before input to Video-LLMs. While efficient, naive strategies such as uniform sampling ignore query information, and similarity-based retrieval with VLMs like CLIP (Radford et al., 2021) improves query alignment but evaluates frames independently (Tang et al., 2025; Liu et al., 2025a), thereby overlooking temporal dependencies and contextual reasoning that are essential for evidence completeness.

In this light, we advance frame selection techniques for long video understanding, by developing a **Se**mantic-**Vi**sual **C**onsensus **E**vidence **S**election (**SeViCES**) framework. SeViCES is training-free and model-agnostic, making it broadly applicable across different Video-LLMs. Our core insight is that long video understanding requires both identifying query-relevant and evidence-complete frames from complementary semantic and visual perspectives, and reconciling these perspectives into a consistent consensus. SeViCES enforces consensus at two levels: (1) **frame level**, where semantic and visual cues jointly guide selection, and (2) **answer level**, where discrepancies between Video-LLM's outputs trigger adaptive refinement of both the selected frames and candidate answers.

To achieve this, SeViCES introduces two key innovations. First, the Semantic–Visual Consensus Frame Selection (SVCFS) module performs dual-perspective selection: (a) a Temporal-Aware Semantic Frame Selection (TAS-FS) branch that uses LLM scoring of captions with temporal context to capture semantic reasoning cues, and (b) a Cluster-guided Mutual Information Frame Selection (CgMI-FS) branch that aligns visual embeddings with semantic scores to ensure representativeness and diversity. Second, the Answer Consensus Refinement (ACR) module compares the answers produced from the two frame sets and resolves inconsistencies through evidence fusion and constrained decoding. Together, these components enable SeViCES to ensure that the final evidence set is both semantically and visually coherent, enabling accurate and consistent reasoning over long videos.

We summarize our contributions as follows:

- We propose SeViCES, a training-free, model-agnostic framework that unifies semantic and visual evidence for long video reasoning and enforces consensus at both frame and answer levels. Across four benchmarks, SeViCES consistently improves multiple Video-LLMs, achieving average gains of around 4.0% and up to 8.3% at best.
- We introduce two novel strategies for query-relevant and evidence-complete frame selection: Temporal-Aware Semantic Frame Selection based on LLM scoring of captions with temporal context, and Cluster-guided Mutual Information Frame Selection based on embedding–score alignment.
- We design an Answer Consensus Refinement module that explicitly uses inconsistencies between semantic- and visual-based predictions as signals to refine evidence and enforce robust consensus.

## 2 RELATED WORKS

**Token selection.** Token selection typically relies on indicators such as attention scores, semantic similarity, and contextual relevance to identify key tokens, thereby reducing the computational cost of large models. FrameFusion (Fu et al., 2024) first merges tokens with cosine similarity above a specified threshold at shallow layers, then calculates cumulative attention scores and applies top-k importance pruning, effectively reducing the number of tokens. PruneVid (Huang et al., 2025) proposes to prune visual tokens through LLM attention computation, achieving substantial reductions in computational cost while preserving task performance. KVTP (Liu et al., 2025c) proposes a query-frame relevance predictor, which is trained to learn the contribution of frame pairs to the task, thereby dynamically determining the token pruning rate for each frame.

**Frame selection.** Early video-language models typically relied on uniform or fixed-interval frame sampling to construct visual inputs. Video-ChatGPT (Maaz et al., 2024), LLaMA-VID (Li et al., 2024) and LongVLM (Weng et al., 2024) adopt evenly spaced frames as visual evidence. While simple and efficient, such strategies often miss fine-grained or short-lived events, motivating the development of adaptive or query-guided frame selection methods.

BOLT (Liu et al., 2025a) proposes leveraging inverse transform sampling to select query-relevant frames to improve VQA performance without requiring model retraining. MDP$^3$ (Sun et al., 2025) combines query-conditioned determinantal point processes with a Markov decision process to efficiently achieve diverse, query-relevant, and sequential frame selection. Frame-Voyager (Yu et al., 2025) extends frame selection by training on frame combinations rather than isolated frames, using prediction losses from a pretrained Video-LLM to rank candidate subsets, thereby capturing inter-frame interactions and temporal complementarities. DynFocus (Han et al., 2025) introduces a mechanism for dynamic event prototype estimation and compact cooperative encoding. It first distinguishes important frames from non-important ones, then applies fine-grained or coarse-grained

encoding accordingly. Two-stage training enables the modules to identify representative frames and compress redundant ones while preserving critical details and temporal structure.

**Comparison with our work.** Unlike prior methods that focus solely on token compression or frame subset selection, our approach is the first to introduce an evidence consensus perspective. Specifically, SeViCES integrates both semantic and visual signals to achieve query-relevant and evidence-complete frame selection, and further enforces answer-level consensus refinement by reconciling discrepancies between semantic- and visual-based reasoning. This dual-level design enables SeViCES to move beyond independent token or frame pruning, establishing a principled consensus-driven framework for reliable long video reasoning.

## 3 METHODOLOGY

The overall framework of our SeViCES is illustrated in Figure 1. It consists of two key components: Semantic-Visual Consensus Frame Selection (SVCFS) and Answer Consensus Refinement (ACR). In the first stage, SVCFS adopts a dual-branch design that integrates complementary semantic and visual cues to identify frames that are both query-relevant and evidence-complete, thereby constructing a reliable evidence pool for reasoning. In the second stage, ACR leverages the responses of an MLLM conditioned on the two frame sets. By explicitly analyzing discrepancies between the semantic-based and visual-based answers, ACR adaptively refines the selected frames and constrains the prediction space to reconcile conflicts. This two-stage design distinguishes SeViCES from prior frame selection methods: instead of relying on unimodal evidence or independent scoring, it establishes a consensus mechanism at both the frame and answer levels. The following sections detail each component in turn.

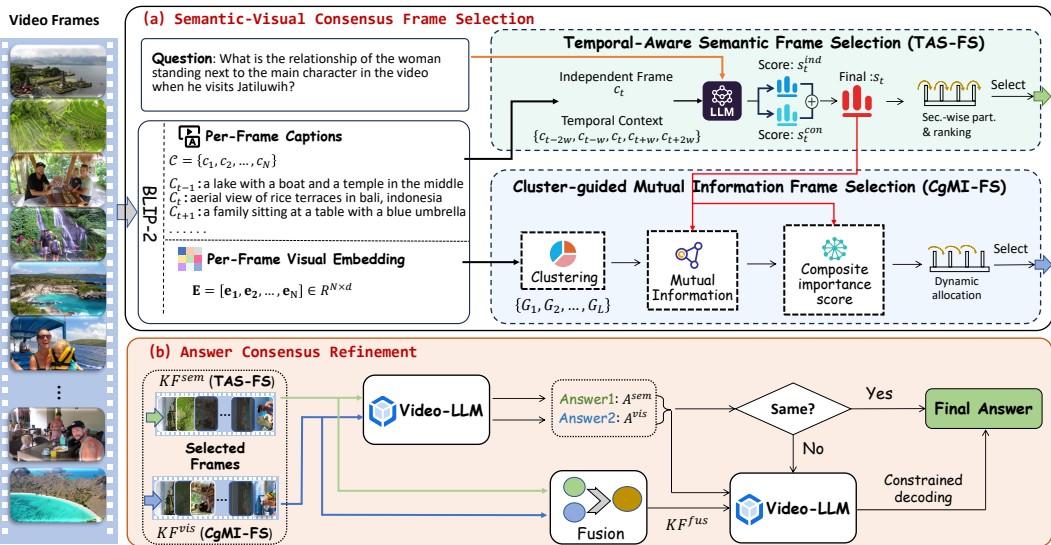

Figure 1: The overall framework of our **SeViCES**. Video frames are processed by BLIP-2 to obtain captions and embeddings. TAS-FS selects query-relevant frames from captions, while CgMI-FS selects frames from embeddings. The two sets are then refined by the ACR module through evidence fusion and constrained decoding to produce the final answer.

### 3.1 SEMANTIC-VISUAL CONSENSUS FRAME SELECTION

Given a video $V = f_1, \cdots, f_{t-1}, f_t, f_{t+1}, \cdots, f_N$ with $N$ frames, the goal of SVCFS is to identify a compact subset of frames (significantly fewer than $N$) that jointly leverage semantic and visual cues while preserving a complete and focused chain of evidence for answering a given question. To this end, we design SVCFS as a dual-branch module: the semantic branch selects frames from the perspective of semantic evidence, while the visual branch selects frames from the perspective of visual evidence. Concretely, these correspond to two complementary strategies: **Temporal-Aware Semantic Frame Selection** (TAS-FS) and **Cluster-guided Mutual Information Frame Selection** (CgMI-FS), respectively.

### 3.1.1 Temporal-Aware Semantic Frame Selection

As discussed in prior works (Tang et al., 2025; Liu et al., 2025a), methods that rely on vision–language models (VLMs) such as CLIP to compute similarity scores between the question and frame features often struggle to identify strongly query-relevant frames. This limitation arises because CLIP-based similarity primarily captures surface-level alignment between visual content and declarative sentences, while effective frame selection for video question answering requires not only semantic matching but also contextual reasoning. For example, given the question "*What does the yellow turtle monster do after receiving a red book?*", the relevant evidence may not only include the frame where the monster is holding the book, but also subsequent frames showing its follow-up actions. Such frames may not be semantically identical to the query description but are nevertheless essential for constructing the complete reasoning chain.

To overcome this limitation, we replace CLIP-based similarity with an LLM-based relevance scoring strategy, leveraging the reasoning and information-matching capabilities of LLMs (e.g., Qwen2-7B-Instruct (Team, 2024)). Specifically, we first use a VLM (e.g., BLIP-2 (Li et al., 2023)) to generate a natural language caption $c_t$ for each frame $f_t$ as follows

$$\{c_1, c_2, \cdots, c_N\}, [\boldsymbol{e}_1, \boldsymbol{e}_2, \cdots, \boldsymbol{e}_N] = \text{BLIP-2}(f_1, f_2, \cdots, f_N), \quad (1)$$

where the embeddings $\mathbf{E}$ are later consumed by the visual branch. These captions are then evaluated by an LLM to produce a semantic relevance score with respect to the question $Q$. To capture both local detail and contextual reasoning, we design two complementary scoring strategies:

$$s_i^{ind} = \text{LLM}(Q, c_t; \text{prompt}_{ind}), \quad (2)$$

$$s_i^{con} = \text{LLM}(Q, \{c_{t-2w}, c_{t-w}, c_t, c_{t+w}, c_{t+2w}\}; \text{prompt}_{con}), \quad (3)$$

where $\text{prompt}_{ind}$ and $\text{prompt}_{con}$ are the instruction prompts for the two modes, respectively. The scores are required to be in the range $[0, 10]$. The frame-independent scoring evaluates each frame solely from its own caption, while the temporal-context scoring incorporates neighboring captions within a stride-$w$ window, enabling the LLM to reason about causal dependencies, abstract concepts, and temporal transitions. Finally, the semantic relevance score of each frame is obtained by combining the two scoring strategies:

$$s_t = s_t^{ind} + s_t^{con}, \quad (4)$$

which balances fine-grained local alignment with broader temporal reasoning, ensuring both query relevance and evidence completeness.

After computing the semantic scores for all frames, a straightforward approach is to directly select the top-$M$ frames with the highest scores. However, such ranking may overconcentrate selections in certain segments, missing critical evidence elsewhere. To mitigate this, we propose a hybrid selection strategy combining global ranking with section-wise partitioning. Specifically, the video is first divided into four equal temporal sections. Within each section, we select the top-$P$ frames ($P < \frac{M}{4}$) by score to guarantee local evidence coverage. From the remaining frames across the entire video, we then select the top-$(M - 4P)$ frames globally. This two-step procedure ensures both temporal diversity and global relevance, thereby improving the completeness of the selected evidence.

### 3.1.2 Cluster-Guided Mutual Information Frame Selection

Visual embeddings provide another crucial modality for video representation, since frame captions alone cannot fully capture the richness of visual content. To complement the semantic branch, we introduce a Cluster-guided Mutual Information Frame Selection (CgMI-FS), which leverages frame embeddings $\mathbf{E} \in \mathbb{R}^{N \times d}$ to incorporate visual evidence.

Specifically, we first cluster the BLIP-2-generated embeddings of all $N$ frames to obtain groups of visually related frames. For clustering, we adopt an improved Density Peaks Clustering (DPC) algorithm (Du et al., 2016) enhanced with K-Nearest Neighbor (KNN) distance computation, which partitions the frame embeddings into compact groups in an unsupervised manner. Each cluster center represents a local mode of the global content distribution, and the distance between a frame embedding $\mathbf{E}_i$ and its corresponding cluster center provides a measure of membership strength. As

a result, the $N$ frames are partitioned into $L$ clusters $G_1, G_2, \cdots, G_L$, which capture the coarse-grained distribution of visual content across the video.

However, clustering alone does not account for the query and thus cannot directly capture relevance. To address this, we further integrate semantic-based relevance information (from LLM-scored captions) into the clustered structure via a mutual information strategy, which aligns clusters with the question and highlights visually representative frames that contribute to aligns-relevant evidence. Concretely, for cluster $G_i$ with $N^i$ frames, let its embeddings be $\mathbf{E}^i$ and their semantic scores be $\mathbf{s}^i = [s_1^i, s_2^i, \cdots, s_{N^i}^i]$. We first reduce $\mathbf{E}^i$ to $\hat{\mathbf{E}}^i \in \mathbb{R}^{N^i \times d'}$ using PCA, preserving $95\%$ of the variance. We then compute the mutual information (Kraskov et al., 2004) between each reduced embedding dimension and the semantic scores:

$$MI_j^i = \text{Mutual-Information}(\hat{\mathbf{E}}_{\cdot|j}^i, \mathbf{s}^i), \quad MI^i = \sum_{j \in d'} MI_j^i. \tag{5}$$

Next, we define a composite importance score $IM$ for each cluster $i$:

$$IM^i = (1 + MI^i) \cdot \bar{s} \cdot \epsilon + \frac{\sigma^2}{2}, \text{where } \epsilon = 1 - \frac{1}{N^i} \sum_{j \in N^i} \text{sim}(\mathbf{e}_i^{center}, \mathbf{e}_j^{center}) \tag{6}$$

$\bar{s}$ is the average semantic score within the cluster, $\sigma^2$ is the score variance, and $\epsilon$ measures the distinctiveness of cluster $i$ with respect to others. This composite score integrates query relevance, intra-cluster diversity, and inter-cluster distinctiveness. Finally, we perform dynamic allocation of the target frame budget (e.g., $M = 64$) to clusters based on their importance scores:

$$M^i = \text{Round}\left( M \times \frac{IM^i}{\sum_j IM^j} \right). \tag{7}$$

For each cluster $i$, the top-$M^i$ frames are then selected according to their semantic scores $s$.

This multi-level selection strategy ensures that the final keyframe set balances query relevance, visual representativeness, and contextual diversity, thereby providing a high-quality evidence pool for long video question answering.

## 3.2 ANSWER CONSENSUS REFINEMENT

SVCFS produces two sets of keyframes, $KF^{sem}$ and $KF^{vis}$, which reflect complementary semantic and visual perspectives of the video. Although both are designed to capture question-relevant evidence, they may emphasize different aspects of the content. Instead of directly fusing these sets at the frame level, we propose an Answer Consensus Refinement (ACR) strategy that evaluates the answers produced by Video-LLMs when conditioned on each set, and uses their agreement or disagreement as a signal for refinement. The central idea is to treat answer inconsistency as evidence incompleteness, and to resolve it through targeted reasoning.

Formally, given a question $Q$, we obtain two initial answers by inputting each keyframe set into the MLLM:

$$A^{sem} = \text{Video-LLM}(Q; KF^{sem}), \quad A^{vis} = \text{Video-LLM}(Q; KF^{vis}). \tag{8}$$

For multiple-choice questions (as in many benchmarks), we assess their consistency by checking whether the predicted option indices coincide. If $A^{sem} = A^{vis}$, we regard the agreement as a sign of high confidence and accept the answer directly.

When the two answers diverge, ACR initiates a refinement process consisting of two steps: evidence fusion and candidate-restricted inference.

**Evidence fusion**. We construct a fused set of keyframes $KF^{fus}$ by taking the union of $KF^{sem}$ and $KF^{vis}$:

$$KF^{fus} = KF^{sem} \cup KF^{vis}. \tag{9}$$

This fusion aggregates all unique evidence identified by the two selection strategies, ensuring that the MLLM has access to a richer and more complete context.

**Constrained decoding**. Rather than re-predicting over the entire answer space, we constrain the MLLM to adjudicate between the two conflicting candidates $A^{sem}$ and $A^{vis}$. That is,

$$A = \text{Video-LLM}(Q; KF^{sem} \cup KF^{vis}; \text{Answer candidates:} \{A^{sem}, A^{vis}\}). \tag{10}$$

This restriction forces the model into a comparative reasoning process, explicitly evaluating which answer is better supported by the fused evidence.

By converting disagreement into a constructive signal, ACR enforces answer-level consensus, reduces ambiguity, and ensures more accurate and robust predictions for long video understanding.

# 4 EXPERIMENTS

## 4.1 EXPERIMENTAL SETUP AND DETAILS

**Dataset and evaluation.** We conduct our experiment on four long video understanding datasets. MLVU (Zhou et al., 2025) contains nine video categories, including movies, surveillance, and others, with durations ranging from three minutes to two hours. The MLVU dev set includes over 1,100 videos and 2,174 corresponding questions. MLVU's overall score is computed as the average accuracy across all task categories. VideoMME (Fu et al., 2025) consists of short-, medium-, and long-duration videos. Each category contains 300 videos, with three questions per video, resulting in a total of 2,700 questions covering six major domains such as knowledge, film, and sports. LongVideoBench (Wu et al., 2024) provides a validation set with more than 700 videos and 1,337 questions. The questions are categorized into two levels—perception and relation—and further divided into 17 fine-grained subcategories. LVBench (Wang et al., 2025b) consists of 103 videos with an average duration exceeding one hour, accompanied by 1,549 multiple-choice questions. It is specifically designed to evaluate models' capability to process ultra-long videos.

**Implementation details.** We integrate the proposed method as a plug-and-play module into several open-source MLLMs, namely LLaVA-Video (Zhang et al., 2025d), Qwen2.5-VL (Bai et al., 2025), and InternVL2.5 (Chen et al., 2025), to evaluate its effectiveness. As the baseline, these models uniformly sample 64 frames from each video for inference. In contrast, our method determines the sampling strategy according to video length defined by the benchmarks: for videos shorter than two minutes, we uniformly sample 128 frames; for videos between two and fifteen minutes, we sample at 1 fps; and for longer videos, we directly sample 1,024 frames uniformly. From these initially sampled frames, we then apply our filtering approach to obtain two distinct sets of 64 frame indices for each query.

Table 1: Performance comparison between our SeViCES and existing methods on four datasets.

| Model | Size | Frames | VideoMME | | | MLVU | LongVB | LVBench |
|---|---|---|---|---|---|---|---|---|
| | | | Medium | Long | Overall | M-Avg | Val | Overall |
| *Proprietary Models* | | | | | | | | |
| GPT-4o | — | 384 | 70.3 | 65.3 | 71.9 | 64.6 | **66.7** | **48.9** |
| Gemini-1.5-Pro | — | 1 fps | **74.3** | **67.4** | **75.0** | — | 64.0 | 33.1 |
| *Open-Source VideoLLMs* | | | | | | | | |
| mPLUG-Owl3 | 7B | 128 | 57.7 | 50.1 | 59.3 | 63.7 | 52.1 | 43.5 |
| NVILA | 8B | 256 | 62.2 | 54.8 | 64.2 | 70.1 | 57.7 | — |
| VideoLLaMA3 | 7B | 180 | 63.7 | 54.9 | 66.2 | 73.0 | 59.8 | 45.3 |
| Oryx-1.5 | 32B | 128 | 65.3 | 59.3 | 67.3 | 72.3 | 62.0 | 30.8 |
| Video-XL-Pro | 3B | 240 | — | — | 60.0 | 70.6 | 56.7 | — |
| SF-LLaVA-1.5 | 7B | 128 | — | — | 63.9 | 71.5 | 62.5 | 45.3 |
| LongVU | 7B | 1 fps | — | **59.5** | 60.6 | 65.4 | — | — |
| ViLAMP | 7B | 1 fps | **65.8** | 57.8 | **67.5** | 72.6 | 61.2 | 45.2 |
| Qwen2.5-VL | 7B | 64 | 62.6 | 53.1 | 63.5 | 63.9 | 60.2 | 41.0 |
| + **SeViCES** (Ours) | 7B | 64 | 65.3 ↑2.7 | 56.8 ↑3.7 | 65.5 ↑2.0 | 72.2 ↑8.3 | **63.9** ↑3.7 | 45.4 ↑4.4 |
| LLaVA-Video | 7B | 64 | 62.3 | 53.0 | 64.4 | 67.9 | 58.9 | 43.1 |
| + AKS | 7B | 64 | — | — | 65.3 | — | 62.7 | — |
| + Suo et al. (2025) | 7B | 32 | — | — | 66.5 | **73.4** | 61.4 | 46.1 |
| + **SeViCES** (Ours) | 7B | 64 | 64.7 ↑2.4 | 56.1 ↑3.1 | 65.6 ↑1.2 | 73.1 ↑5.2 | 63.1 ↑4.2 | **47.3** ↑4.2 |
| InternVL2.5 | 8B | 64 | — | 52.8 | 64.2 | 68.7 | 59.3 | 43.4 |
| + Suo et al. (2025) | 8B | 32 | — | — | 65.3 | 70.0 | 60.6 | 46.6 |
| + **SeViCES** (Ours) | 8B | 64 | 63.2 | 55.2 ↑2.4 | 64.7 ↑0.5 | 72.1 ↑3.4 | 61.7 ↑2.4 | 46.7 ↑3.3 |

## 4.2 RESULTS AND ANALYSIS

We evaluate our method by comparing the performance of recent mainstream proprietary and open-source multimodal large language models (MLLMs) across four long video understanding benchmarks. The proprietary models include *GPT-4o* (Hurst et al., 2024) and *Gemini-1.5-Pro* (Team et al., 2024). The open-source models include *mPLUG-Owl3* (Ye et al., 2025), *NVILA* (Liu et al., 2025d), *VideoLLaMA3* (Zhang et al., 2025a), *Oryx-1.5* (Liu et al., 2025e), *Video-XL-Pro* (Liu et al., 2025b), *SF-LLaVA-1.5* (Xu et al., 2025), *LongVU* (Shen et al., 2025b), and *ViLAMP* (Cheng et al., 2025).

### 4.2.1 COMPARISON WITH STATE-OF-THE-ARTS

Table 1 reports the results across the four datasets. We have the following observations:

**Effectiveness and generalization of SeViCES.** When integrated into three different Video-LLMs, SeViCES consistently delivers substantial performance gains across all benchmarks. For instance, on Video-MME, Qwen2.5-VL with SeViCES improves accuracy by 2.7% on medium-duration videos (∼10 minutes) and by 3.7% on long-duration videos (∼40 minutes) compared with uniform 64-frame sampling. On MLVU, LongVB, and LVBench, SeViCES achieves additional gains of 8.3%, 3.7%, and 4.4%, respectively. These results demonstrate that SeViCES is both effective in selecting informative frames and broadly generalizable across different Video-LLMs.

**Comparison with other training-free frame selection methods.** AKS (Tang et al., 2025) and Suo et al. (2025) are representative training-free frame selection methods. Compared with AKS, SeViCES consistently performs better across all datasets with the baseline LLaVA-Video. Against Suo et al. (2025), SeViCES achieves superior results on most benchmarks: it is slightly weaker on Video-MME but obtains consistently higher performance on LongVB and LVBench across both LLaVA-Video and InternVL2.5 backbones. Considering that Suo et al. (2025) heavily relies on repeated LLM and Video-LLM calls, SeViCES achieves these improvements with higher efficiency.

**Comparison with open-source Video-LLMs.** By equipping Video-LLMs with SeViCES, their performance surpasses many open-source models, even when using only 64 frames. For example, Qwen2.5-VL (7B) combined with SeViCES significantly outperforms Oryx-1.5 (32B) on MLVU, LongVB, and LVBench, highlighting the efficiency and scalability advantages of our approach.

### 4.2.2 PERFORMANCE ACROSS VIDEO TASK TYPES

Figure 2 compares the performance of LLaVA-Video and Qwen2.5-VL on the LVBench dataset, both with and without SeViCES, across five distinct video understanding tasks: information retrieval, event understanding, entity recognition, reasoning, and temporal grounding. Notably, on the Key Information Retrieval task, Qwen2.5-VL improves from 39.9% to 50.2% accuracy, showing that SeViCES effectively identifies and preserves frames containing information critical to the query. This highlights the ability of our selection strategy to filter noise while retaining content most relevant for precise retrieval. When averaging results across LLaVA-Video and Qwen2.5-VL, we observe gains of 5.5% on the Entity Recognition task and 3.2% on the Event Understanding task. These improvements indicate that SeViCES enhances the models' ability to capture key scenes, follow event progression, and track entities, which in turn enables clearer reasoning and more accurate temporal analysis.

## 4.3 ABLATION STUDY

In the ablation study, we examine the effects of SeViCES components and some hyper-parameters using Qwen2.5-VL baseline and VideoMME dataset.

**Effect of key components of SeViCES.** SeViCES consists of two main components: Semantic–Visual Consensus Frame Selection (SVCFS) and Answer Consensus Refinement (ACR). SVCFS integrates two complementary strategies, i.e., Temporal-Aware Semantic Frame Selection (TAS-FS) and Cluster-guided Mutual Information Frame Selection (CgMI-FS), to identify frames. To evaluate their contributions, we feed the frame sets generated by TAS-FS and CgMI-FS separately into Qwen2.5-VL. As shown in Table 2, both TAS-FS and CgMI-FS yield substantial improvements over the baseline across all settings. In particular, on the long-video setting (average duration ∼40 minutes), TAS-FS improves accuracy by 1.7%, while CgMI-FS achieves a much higher 3.1% gain. This

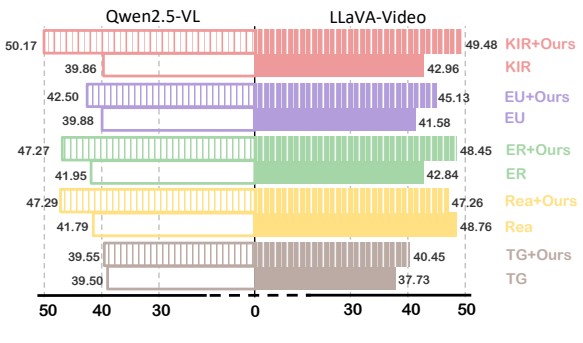

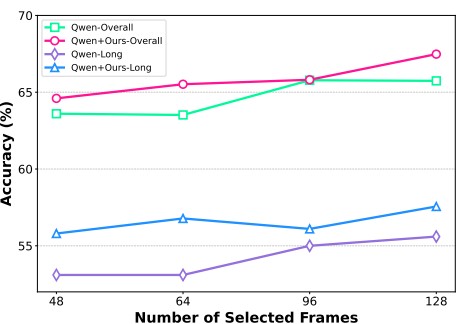

Figure 2: Performance of LLaVA-Video and Qwen2.5-VL on LVBench tasks with and without SeViCES. Task types include Key Information Retrieval (KIR), Event Understanding (EU), Entity Recognition (ER), Reasoning (Rea), and Temporal Grounding (TG).

Figure 3: Performance changes with different numbers of selected frames.

Table 2: Performance changes of different SeViCES components.

| Method | Overall | Medium | Long |
|---|---|---|---|
| Qwen2.5-VL | 63.5 | 62.6 | 53.1 |
| + SVCFS (TAS-FS) | 64.4 ↑0.9 | 64.3 ↑1.7 | 54.8 ↑1.7 |
| + SVCFS (CgMI-FS) | 64.4 ↑0.9 | 63.9 ↑1.3 | 56.2 ↑3.1 |
| + SVCFS+ACR | 65.5 ↑2.0 | 65.3 ↑2.8 | 56.8 ↑3.7 |

Table 3: Performance changes with different number $L$ of clusters.

| Method | Overall | Medium | Long |
|---|---|---|---|
| Qwen2.5-VL | 63.5 | 62.6 | 53.1 |
| $L=8$ | 64.3 | 62.6 | 55.6 |
| $L=12$ | 64.4 | 63.9 | 56.2 |
| $L=16$ | 64.2 | 64.7 | 55.8 |

demonstrates that SVCFS effectively identifies key frames even under challenging long-duration scenarios. Furthermore, incorporating ACR for answer-level refinement brings additional improvements, confirming that consensus-based adjudication helps resolve ambiguities and enhances robustness.

**Effect of the number $M$ of selected frames.** The number of input frames $M$ has a direct impact on answer accuracy. We evaluate Qwen2.5-VL with $M \in 48, 64, 96, 128$. As shown in Figure 3, accuracy generally improves as more frames are provided. When SeViCES is applied, performance increases steadily across all settings, consistently surpassing the baseline. Notably, even in cases where baseline accuracy plateaus or slightly fluctuates with larger $M$, SeViCES maintains clear advantages. These results demonstrate that our semantic–visual consensus strategy scales effectively with frame budget and yields robust improvements over uniform sampling.

**Effect of the number $L$ of clusters in CgMI-FS.** CgMI-FS groups frames via clustering, making the number of clusters $L$ a critical hyper-parameter. We evaluate three settings ($L = 8, 12, 16$) and report their results in Table 3. Two key observations emerge: (1) clustering consistently improves performance over the baseline, regardless of the chosen $L$, confirming the effectiveness of modeling global visual structure; and (2) among the tested values, $L = 12$ achieves the best balance, yielding superior results on both the long-video and overall settings. Therefore, we adopt $L = 12$ as the default configuration in the experiments.

### 4.4 EXAMPLE DEMONSTRATION

To visually demonstrate the complementary strengths of the semantic- and visual-based frame selection strategies (TAS-FS and CgMI-FS), as well as the effectiveness of the answer refinement module (ACR), we present two representative video QA cases: (1) a single-correct multiple-choice question and (2) an elimination-type multiple-choice question.

For the single-correct question (Figure 4(1)), the query is: "*What does the yellow turtle monster do after receiving a red book?*". The critical cues include both the "*yellow turtle monster*" and the "*red book*", and the answer depends on selecting frames that occur "*after the book is received*". In

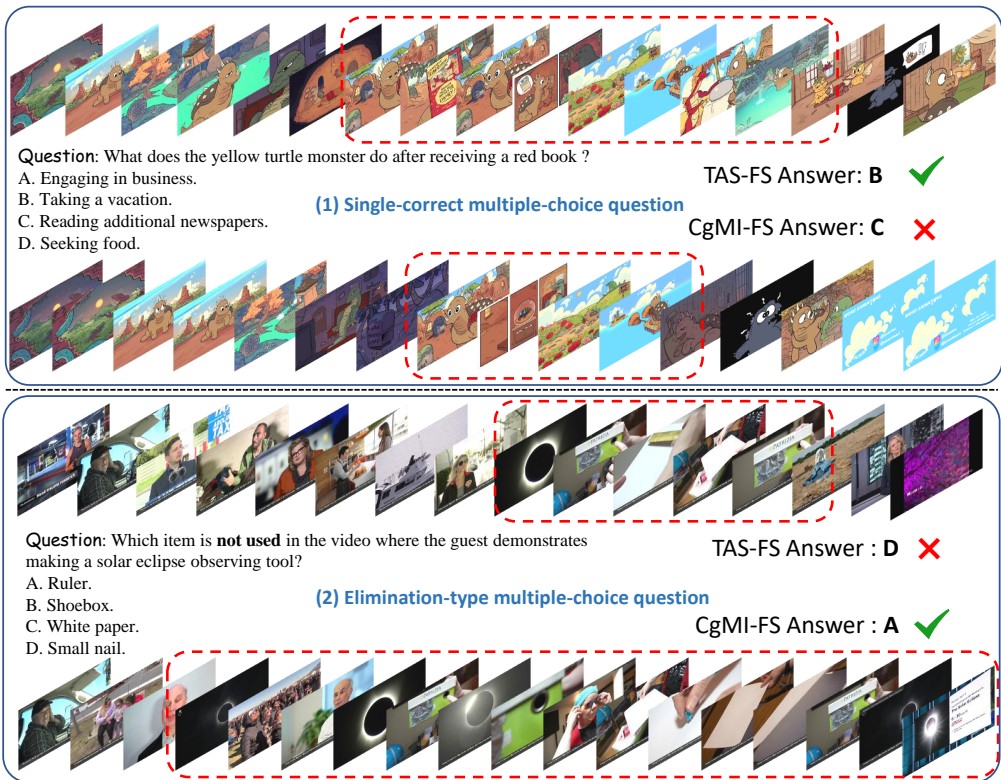

Figure 4: Two representative VideoQA task examples using Qwen2.5-VL and VideoMME dataset.

practice, the LLM-based TAS-FS successfully identifies frames capturing this temporal dependency, highlighting both the object and its subsequent action. In contrast, the clustering-based CgMI-FS primarily selects frames containing the yellow turtle monster but often overlooks the red book, likely because the book appears only briefly and co-occurs with the monster.

For the elimination-type question (Figure 4(2)), the query is: "*Which item is not used in the video where the guest demonstrates making a solar eclipse observing tool?*". This task requires identifying all relevant objects in the video. Here, the clustering-based CgMI-FS, which models global visual distributions, excels at capturing a diverse set of objects, making it more effective than TAS-FS alone.

Finally, when the frame sets from TAS-FS and CgMI-FS are fused, the ACR module resolves their discrepancies and guides the Video-LLM to the correct final answer. These case studies illustrate how TAS-FS and CgMI-FS provide complementary evidence, i.e., semantic reasoning for temporal dependencies and visual clustering for global coverage, and how ACR integrates them into a robust consensus.

## 5 CONCLUSION

We have introduced SeViCES, a training-free and model-agnostic framework for long video understanding with Video-LLMs. By combining semantic–visual consensus frame selection (SVCFS) and answer consensus refinement (ACR), SeViCES ensures that selected frames are both query-relevant and evidence-complete, while inconsistencies between semantic and visual reasoning are turned into constructive signals for refinement. Experiments across multiple benchmarks show that SeViCES consistently boosts accuracy and robustness over strong baselines and existing training-free methods, even with limited frame budgets. Our analysis further demonstrates the scalability of the approach and its generalizability across different Video-LLMs. These results highlight the promise of consensus-driven evidence selection as a principled strategy for improving the reliability of long video understanding.

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

## A APPENDIX

In this appendix, we provide additional implementation details, additional results, and more ablation studies. For the use of large language models, we use LLMs only for language polishing.

## A.1 Additional Implementation Details

We firstly provide additional implementation details. We use the Qwen2-7B-Instruct model to score the relevance between each frame's caption and the question text. Prompts for LLM:

```
prompt_ind = (
f"You are an expert in image-text matching.\n"
f"Question: {question}\n"
f"Image Caption: {caption}\n"
f"Please evaluate the relevance between the image caption and the
    question by giving a score from 0 to 10, "
f"where 10 means highly relevant and 0 means completely
    irrelevant. Please respond with a single number only, without
    explanation."
)

prompt_con = (
"You are an expert in understanding events in long videos.\n"
"Below is a sequence of frames (described with captions), listed
    in the order they appear in the video.\n"
"You are given a question about the video. Please judge how
    relevant the **middle frame** is to the question.\n"
"Give a score between 0 (not relevant at all) and 10 (highly
    relevant).\n\n"
f"Question: {question}\n\n"
f"Frame sequence:\n"
f"{context_text}\n\n"
f"The middle frame is Frame {len(prev_caps)+1}.\n"
f"How relevant is this frame to the question?\n"
f"Answer with a single number (0-10):"
)
```

We use the LMMs-Eval (Zhang et al., 2025b) evaluation suite to test several Video-LLMs on different benchmarks.

Table 4: Results of our SeViCES with the three Video-LLMs on four datasets.

| Model | VideoMME | | | MLVU | LongVB | LVBench |
|---|---|---|---|---|---|---|
| | Medium | Long | Overall | M-Avg | Val | Overall |
| Qwen2.5-VL-7B | 62.6 | 53.1 | 63.5 | 63.9 | 60.2 | 41.0 |
| + SVCFS (TAS-FS) | 64.3 ↑1.7 | 54.8 ↑1.7 | 64.4 ↑0.9 | 69.3 ↑5.4 | 62.0 ↑1.8 | 44.9 ↑3.9 |
| + SVCFS (CgMI-FS) | 63.9 ↑1.3 | 56.2 ↑3.1 | 64.4 ↑0.9 | 71.4 ↑7.5 | 62.6 ↑2.4 | 44.7 ↑3.7 |
| + SeViCES (SVCFS+ACR) | 65.3 ↑2.7 | 56.8 ↑3.7 | 65.5 ↑2.0 | 72.2 ↑8.3 | 63.9 ↑3.7 | 45.4 ↑4.4 |
| LLaVA-Video-7B | 62.3 | 53.0 | 64.4 | 67.9 | 58.9 | 43.1 |
| + SVCFS (TAS-FS) | 62.9 ↑0.6 | 55.0 ↑2.0 | 64.8 ↑0.4 | 71.0 ↑3.1 | 61.8 ↑2.9 | 46.4 ↑3.3 |
| + SVCFS (CgMI-FS) | 63.2 ↑0.9 | 55.8 ↑2.8 | 64.5 ↑0.1 | 71.6 ↑3.7 | 61.5 ↑2.6 | 46.3 ↑3.2 |
| + SeViCES (SVCFS+ACR) | 64.7 ↑2.4 | 56.1 ↑3.1 | 65.6 ↑1.2 | 73.1 ↑5.2 | 63.1 ↑4.2 | 47.3 ↑4.2 |
| InternVL2.5-8B | — | 52.8 | 64.2 | 68.7 | 59.3 | 43.4 |
| + SVCFS (TAS-FS) | 62.4 | 55.2 ↑2.4 | 64.4 ↑0.2 | 72.0 ↑3.3 | 60.9 ↑1.6 | 46.5 ↑3.1 |
| + SVCFS (CgMI-FS) | 62.7 | 54.2 ↑1.4 | 64.2 | 71.5 ↑2.8 | 59.8 ↑0.5 | 45.0 ↑1.6 |
| + SeViCES (SVCFS+ACR) | 63.2 | 55.2 ↑2.4 | 64.7 ↑0.5 | 72.1 ↑3.4 | 61.7 ↑2.4 | 46.7 ↑3.3 |

## A.2 Additional Results

Then, we provide additional results of SeViCES evaluated with three different Video-LLMs across four long video understanding datasets, as shown in Table 4. We report detailed performance for each component of our framework, including the Temporal-Aware Semantic Frame Selection (TAS-FS), Cluster-Guided Mutual Information Frame Selection (CgMI-FS), and the Answer Consensus

Refinement (ACR) module. These results further validate the contribution of each component and highlight the robustness of SeViCES across diverse settings.

## A.3 MORE ABLATION STUDIES

We further conduct ablation studies on the design choices of the semantic-based frame selection strategy (TAS-FS). As shown in Table 5, we first compare two LLM scoring strategies: frame-independent scoring and temporal-context scoring. The results show that temporal-context scoring slightly outperforms frame-independent scoring, particularly on medium-length videos, while the two perform comparably on very long videos. We speculate that this is because the time window used ($w = 5$, i.e., 5 seconds) remains relatively short compared to the duration of very long videos ($\sim$40 minutes).

Second, we examine the effect of combining the two scoring strategies. The results in the last two rows indicate that the combination of frame-independent and temporal-context scoring substantially improves performance over either strategy alone. Finally, incorporating the section-wise selection scheme yields the best overall performance, confirming the effectiveness of ensuring both local coverage and global relevance in semantic frame selection.

Table 5: Performance changes with different TAS-FS design strategies on VideoMME dataset.

| Method | Overall | Medium | Long |
|---|---|---|---|
| Qwen2.5-VL | 63.5 | 62.6 | 53.1 |
| + TAS-FS (frame-independent scoring) | 63.5 | 61.7 | 54.4 |
| + TAS-FS (temporal-context scoring) | 63.8 | 63.3 | 54.3 |
| + TAS-FS (rank scores and select top-64) | 64.3 | 63.7 | 55.2 |
| + TAS-FS (section-wise selection, used) | 64.4 | 64.3 | 54.8 |

