# OpenReview forum: "SeViCES: Unifying Semantic-Visual Evidence Consensus for Long Video Understanding"
_ICLR.cc/2026/Conference — ICLR 2026 Conference Withdrawn Submission_

### Official Review · Reviewer_Qim2 · 2025-10-29

**Soundness:** 3
**Presentation:** 3
**Contribution:** 3
**Rating:** 4
**Confidence:** 4

**Summary:**

This paper introduces SeViCES, a training-free and model-agnostic framework that enhances long video understanding by selecting key frames through a semantic-visual consensus mechanism and refining final answers by resolving inconsistencies between different evidence sources.

**Strengths:**

- A significant strength is the proposed dual-branch (semantic and visual) frame selection module, which explicitly addresses the limitation of unimodal approaches by leveraging the complementary strengths of LLM-based reasoning on captions and cluster-guided visual alignment to capture more complete, query-relevant context.

- The paper is well-written and easy to follow.

**Weaknesses:**

1.  Regarding the "Semantic-Visual Consensus Frame Selection" method, the authors argue that traditional CLIP-based methods for measuring text-frame relevance are difficult to apply directly to videos containing temporal information. They instead propose converting frames into captions and using an LLM for assessment. I have the following concerns:
    - Could the frame-to-caption conversion process itself lead to inaccurate descriptions due to the loss of temporal information?

    - Since the individual frame caption (s_ind) still lacks temporal context, is using an LLM here fundamentally more advantageous than using CLIP?

    - For the consensus caption (s_con), which incorporates contextual captions, is feeding this textual window into the LLM indeed superior to using a visual window of the corresponding frames (e.g., with a video encoder like VideoCLIP) for relevance calculation? Intuitively, a window composed of single-frame captions likely loses more temporal information compared to a window of the original visual frames.

2.  In Table 1, while the method is compared against numerous video-LLMs, it is only benchmarked against two key frame selection methods. This is insufficient to demonstrate the advantage of SeViCES. Notably, compared to Suo et al. (2025), the improvements are not always significant. Furthermore, the comparison seems unfair as Suo et al. (2025) uses only 32 frames. I strongly recommend including more comparisons with state-of-the-art frame selection methods under fair conditions, such as [1, 2].

[1] Q-Frame: Query-aware Frame Selection and Multi-Resolution Adaptation for Video-LLMs

[2] ViaRL: Adaptive Temporal Grounding via Visual Iterated Amplification Reinforcement Learning


3.  The paper lacks sufficient ablation studies for the "Answer Consensus Refinement" component. Specifically, the individual contribution and effectiveness of the "Evidence fusion" and "Constrained decoding" techniques should be validated through experiments.

**Questions:**

See Weaknesses

---

> ### Author Response · Authors · 2025-11-21
> **Response to Reviewer Qim2**
>
> Thank you very much for your recognition of the methodology in our paper.
>
> > Could the frame-to-caption conversion process itself lead to inaccurate descriptions due to the loss of temporal information?
>
> Precisely because relying on individual frames alone lacks temporal context, we adopt a **temporal-context scoring** strategy. This prompts the LLM to jointly consider the key content from neighboring frames' captions and holistically assess the importance of each frame.
>
> > is using an LLM here fundamentally more advantageous than using CLIP?
>
> | Method           | Long (%) | Overall (%) |
> |------------------|----------|-------------|
> | CLIP (Top-64)    | 53.0     | 63.9        |
> | TAS-FS (Top-64)  | 54.2     | 64.5       |
>
> This table directly compares the performance difference between using CLIP and TAS-FS to compute frame relevance—both approaches simply select the top-64 highest-scoring frames from the video. The results are evaluated using the LLaVA-Video model on the **VideoMME** benchmark.
>
> > is feeding this textual window into the LLM indeed superior to using a visual window of the corresponding frames (e.g., with a video encoder like VideoCLIP) for relevance calculation?
>
> We are currently attempting to reproduce VideoCLIP or identify a more general model that effectively incorporates temporal information in videos, in order to compare it with our method.
>
> > the comparison seems unfair as Suo et al. (2025) uses only 32 frames.
>
> Suo et al. (2025) first sample **32 frames randomly from the video for each attempt**, repeat this **10 times**, and then perform additional processing to answer one question. So in total they use **at least 320 frames**. We directly kept the “32” from their original table, which caused the misunderstanding. We sincerely apologize for this oversight in the details.
>
> _Suo et al. From Trial to Triumph: Advancing Long Video Understanding via Visual Context Sample Scaling and Self-reward Alignment. ICCV, 2025._
>
>
> > I strongly recommend including more comparisons with state-of-the-art frame selection methods under fair conditions, such as [1, 2].
>
> | Method                              | Frames (Resolution)                     | Short (%) | Medium (%) | Long (%) | Overall (%) |
> |-------------------------------------|------------------------------------------|-----------|------------|----------|-------------|
> | Qwen2.5-VL + Q-Frame                | 4 + 8 + 32 (high + medium + low)         | 70.1      | 56.7       | 49.6     | 58.8        |
> | Qwen2.5-VL + Q-Frame                | 16 + 16 + 32                             | 71.9      | 59.1       | 52.1     | 61.04       |
> | Qwen2.5-VL + SEViCES (Ours)                | 44 low-resolution frames                 | 74.1      | 60.6       | 55.1     | 63.26       |
>
> We successfully reproduced Q-Frame and compared it with our method. Q-Frame uses a relatively small number of frames overall and employs multiple resolutions. We experimented with various combinations of frame resolutions in Q-Frame, but all configurations yielded results inferior to our method.
>
> **ViaRL** introduces rule-based reinforcement learning for video temporal localization and employs an alternating training procedure. Their approach is already resource-intensive, requiring model training. In contrast, our method is purely training-free for frame selection and cannot be directly compared with theirs. Moreover, this paper does not compare with other frame selection methods, and only evaluates against baselines of other models on a single open-source model.

---

### Official Review · Reviewer_YSsL · 2025-10-31

**Soundness:** 3
**Presentation:** 2
**Contribution:** 2
**Rating:** 2
**Confidence:** 4

**Summary:**

The paper proposes SeViCES, a training-free, model-agnostic framework for long video understanding that unifies semantic and visual evidence through two main modules: Semantic–Visual Consensus Frame Selection (SVCFS) and Answer Consensus Refinement (ACR). SeViCES effectively selects query-relevant and evidence-complete frames, achieving significant performance improvements across multiple VideoLLMs and benchmarks.

**Strengths:**

1. **Training-free and model-agnostic design.** The proposed method can be plugged into multiple VideoLLMs without training.
2. **Consistent performance gains.** SeViCES shows consistent performance gains across multiple video benchmarks, including long video understanding tasks, showing its effectiveness.

**Weaknesses:**

1. **Limited novelty.** The proposed SeViCES framework shows limited novelty. The core ideas, (1) LLM-based frame caption scoring and (2) visual feature-based frame clustering for frame selection, have already been explored in VideoTree [1]. As such, both the objective and methodology of SeViCES closely resemble those of VideoTree. A more detailed performance comparison and discussion of their differences and advantages are needed to justify the novelty of this work.

    [1] Wang et al., VideoTree: Adaptive Tree-based Video Representation for LLM Reasoning on Long Videos, CVPR 2025

2. **Computational overhead.** The proposed framework introduces substantial computational complexity due to multiple caption generations, LLM-based scoring, kNN-based clustering, and related steps. The paper should include a quantitative analysis and discussion of computational overhead (e.g., throughput, latency, or runtime per video) to demonstrate practical feasibility.
3. **Missing ablations.** The dynamic frame allocation mechanism (Eq. 7) appears overly complex, yet its contribution is not clearly analyzed. It is necessary to perform ablation experiments to verify whether this equation is indeed critical. How would the performance change if frames were allocated equally across clusters instead?
4. **Unclear writing.** Some notations are not clearly defined.
    1. Precise definition of Mutual-Information (Eq. 5) and sim() (Eq.6) are missing
    2. In Eq. 6, superscript $i$ seems to be missing for $\bar{s}$, $\sigma^2$, and $\epsilon$.

**Questions:**

Please refer to weaknesses.

---

> ### Author Response · Authors · 2025-11-21
> **Response to Reviewer YSsL (Part 1/2)**
>
> > 1. Limited novelty.
>
> | Aspect               | VideoTree      | SEViCES                     |
> |----------------------|----------------|-----------------------------|
> | Relevance Scoring Model    | GPT-4o         | Qwen                        |
> | Scoring Strategy     | Only cluster centers | fame-independent scoring $s_i^{\text{ind}}$ + temporal-context scoring $s_i^{\text{con}}$ |
> | Clustering Method    | K-means        | DPC-KNN + Mutual Information (multimodal) |
> | Final Reasoning Input | Captions of key frames → GPT-4o | Key frames directly → Open-source MLLM     |
> | Post-hoc Refinement  | None           | ACR Module                  |
>
> VideoTree uses GPT-4o to score the 8 cluster centers with ratings of 1, 2, or 3. Clusters with higher scores are retained and further subdivided into two second-level clusters, enabling iterative clustering refinement.
>
> Unlike methods that select frames solely based on the caption scores of individual cluster-center frames, our approach incorporates contextual information from surrounding frames when computing relevance scores. After a single round of clustering, we compute the mutual information between each frame’s visual features and its semantic content to measure their relevance. Key frames are then dynamically allocated based on this metric.
>
> Additionally, the ACR module performs post-hoc adjustment to mitigate answer fluctuations caused by minor variations in the input frames.
>
> > 2. Computational overhead.
>
> We adaptively select the number of sampled frames based on video duration.
> BLIP-2 takes an average of 0.09 seconds per frame to generate captions and embeddings for videos. When scoring each frame individually, we employ a binning strategy to select multiple captions from frames that are sufficiently distant from one another, and input them together into the LLM. The LLM then outputs the scores for all selected frames in a single response formatted as a list. This approach avoids the need to invoke the LLM repeatedly for each individual frame.
>
> For fewer than 15% of videos on average, the corresponding questions require an additional inference step through constrained decoding in the Answer Consensus Refinement module.
>
> **Table 1: Evaluation Computational Cost on VideoMME Benchmark**
> | Method   | Total Frames Input to MLLM/LLM for Frame Selection | Number of MLLM Calls |
> |----------|----------------------------------------------------|----------------------|
> | DVD      | 5,544,000 (MLLM)                                         | 589,500              |
> | Vgent    | 2,772,000 (MLLM)                                         | 136,800              |
> | Suo et al.   | 1,144,800 (MLLM)                                         | 38,340               |
> | SeViCES (Ours) | 1,522,800 (LLM)                          | 5,700                |
>
> Treating MLLMs or LLMs as agents and invoking them multiple times within a method has become increasingly common in the field of long-form video understanding [4]. We compare the computational costs of several recent methods in **Table 1**.
>
> DVD [1] divides videos into 5-second segments and samples each segment at 2 FPS, feeding the frames into GPT-4.1 to generate captions and build a database. During the iterative reasoning loop, DVD employs OpenAI o3 for agent-based reasoning and planning, while using GPT-4.1 to perform visual question answering on a subset of frames.
>
> Vgent [2] first samples long videos at 1 FPS and segments them accordingly. MLLMs (e.g., Qwen2.5-VL) extract key entities, actions, and scenes from both the question and each sampled frame. They further decompose the original question into multiple sub-questions, feed each video segment into the MLLM to answer these sub-questions, and finally input the selected frames into the MLLM to generate the final answer.
>
> Suo et al.[3] uniformly splits long videos into 32 segments and performs bin sampling by randomly selecting one frame from each segment. They generate answers by querying the model 10 times repeatedly. When disagreements arise, they invoke the model an additional 9–11 times (depending on the question type) to produce supplementary answers.
>
> [1] [Deep Video Discovery: Agentic Search with Tool Use for Long-form Video Understanding. NeurIPS. 2025.](https://openreview.net/forum?id=oQYq9L1NVT)
>
> [2] [Vgent: Graph-based Retrieval-Reasoning-Augmented Generation For Long Video Understanding. NeurIPS. 2025.](https://openreview.net/forum?id=5xPvWat3IX)
>
> [3] [From Trial to Triumph: Advancing Long Video Understanding via Visual Context Sample Scaling and Self-reward Alignment. ICCV, 2025.](https://openaccess.thecvf.com/content/ICCV2025/html/Suo_From_Trial_to_Triumph_Advancing_Long_Video_Understanding_via_Visual_ICCV_2025_paper.html)
>
> [4] [Adavideorag: Omni-contextual adaptive retrieval-augmented efficient long video understanding. NeurIPS. 2025.](https://openreview.net/forum?id=FDAI0PY9Qp)

---

> ### Author Response · Authors · 2025-11-21
> **Response to Reviewer YSsL (Part 2/2)**
>
> > 3. Missing ablations.
>
> Thank you for your suggestion. If frames were allocated uniformly across all clusters—i.e., simply picking the top-64 highest-scoring frames from the entire video—it would be equivalent to directly selecting the 64 frames with the highest individual scores. This approach significantly degrades accuracy because it overly concentrates on a narrow subset of frames, missing the broader narrative context and causal relationships essential for correct reasoning.
>
> For reference, performance comparisons are as follows:
>
> | Method               | Short (%) | Medium (%) | Long (%) | Overall (%) |
> |----------------------|-----------|------------|----------|-------------|
> | Direct Top-64        | 76.0      | 62.7       | 55.2     | 64.63       |
> | CgMI-FS              | 76.2      | 63.2       | 55.8     | 65.1        |
>
> These are the results of our evaluation on VideoMME using the LLaVA-Video model.
>
> > 4. Unclear writing.
>
> Thank you for pointing out the issues. sim() (Eq. 6) is the cosine similarity. We will address and refine them in our subsequent revisions.

---

### Official Review · Reviewer_ZkXQ · 2025-11-01

**Soundness:** 2
**Presentation:** 2
**Contribution:** 2
**Rating:** 2
**Confidence:** 4

**Summary:**

This paper proposes SeVICES, a novel training-free and model-agnostic framework to address the challenge of long video understanding. To overcome the prohibitive computational cost of processing all frames, SeVICES selects a compact, query-relevant subset of frames by enforcing consensus at two levels. First, the Semantic-Visual Consensus Frame Selection (SVCFS) module uses two complementary branches: (1) a semantic branch (TAS-FS) that uses an LLM to score frame captions based on query-relevance and temporal context, and (2) a visual branch (CgMI-FS) that uses clustering and mutual information to select a visually diverse set of frames that are also semantically relevant. Second, the Answer Consensus Refinement (ACR) module runs the Video-LLM on both sets of selected frames. If the answers disagree, it treats this as a sign of incomplete evidence, fuses the two frame sets, and runs the Video-LLM a final time, forcing it to adjudicate between the two conflicting answers.

**Strengths:**

- The paper is easy to follow and well-written.
- The proposed method is training-free and model-agnostic, which is easy to extend.

**Weaknesses:**

- The paper is motivated by overcoming the "prohibitive computational costs" of processing long videos. However, the proposed SeVICES framework introduces a new, significant, and entirely unmeasured computational bottleneck: inference latency. So, it would be better to provide the complexity analysis of the proposed method.
- A primary strength of a "training-free" method should be its universal applicability to any MLLM. However, the paper's experiments are limited to three open-source models. The method was not applied to any state-of-the-art proprietary models (e.g., GPT-4V, Gemini), which would have been a powerful demonstration of its model-agnostic claim.
- The core Answer Consensus Refinement (ACR) module is fundamentally incompatible with open-ended tasks, as it is designed only for multiple-choice questions. Could you provide the experimental results of the proposed method on open-ended tasks?
- Experimental results on SVCFS (TAS-FS + FGMI-FS) without ACR are removed from Table 2.

**Questions:**

- Detailed explanation on notation $s_i^{ind}, s_i^{con}$. There is no explanation that $s_i^{ind}$ is a frame-independent score and $s_i^{con}$ is a temporal-context score.
- The paper motivates the "section-wise partitioning" strategy by claiming that a simple top-M score ranking "may overconcentrate selections in certain segments". This is a key justification for the TAS-FS design. Could the authors provide a qualitative example that visualizes this failure case?

---

> ### Author Response · Authors · 2025-11-21
> **Response to Reviewer ZkXQ (Part 1/2)**
>
> Thank you very much for your valuable suggestions. We have conducted some experiments as per your request.
>
> >it would be better to provide the complexity analysis of the proposed method.
>
> We adaptively select the number of sampled frames based on video duration.
> BLIP-2 takes an average of 0.09 seconds per frame to generate captions and embeddings for videos. When scoring each frame individually, we employ a binning strategy to select multiple captions from frames that are sufficiently distant from one another, and input them together into the LLM. The LLM then outputs the scores for all selected frames in a single response formatted as a list. This approach avoids the need to invoke the LLM repeatedly for each individual frame.
>
> For fewer than 15% of videos on average, the corresponding questions require an additional inference step through constrained decoding in the Answer Consensus Refinement module.
>
> **Table 1: Evaluation Computational Cost on VideoMME Benchmark**
> | Method   | Total Frames Input to MLLM/LLM for Frame Selection | Number of MLLM Calls |
> |----------|----------------------------------------------------|----------------------|
> | DVD      | 5,544,000 (MLLM)                                         | 589,500              |
> | Vgent    | 2,772,000 (MLLM)                                         | 136,800              |
> | Suo et al.   | 1,144,800 (MLLM)                                         | 38,340               |
> | SeViCES (Ours) | 1,522,800 (LLM)                          | 5,700                |
>
> Treating MLLMs or LLMs as agents and invoking them multiple times within a method has become increasingly common in the field of long-form video understanding [4]. We compare the computational costs of several recent methods in **Table 1**.
>
> DVD [1] divides videos into 5-second segments and samples each segment at 2 FPS, feeding the frames into GPT-4.1 to generate captions and build a database. During the iterative reasoning loop, DVD employs OpenAI o3 for agent-based reasoning and planning, while using GPT-4.1 to perform visual question answering on a subset of frames.
>
> Vgent [2] first samples long videos at 1 FPS and segments them accordingly. MLLMs (e.g., Qwen2.5-VL) extract key entities, actions, and scenes from both the question and each sampled frame. They further decompose the original question into multiple sub-questions, feed each video segment into the MLLM to answer these sub-questions, and finally input the selected frames into the MLLM to generate the final answer.
>
> Suo et al.[3] uniformly splits long videos into 32 segments and performs bin sampling by randomly selecting one frame from each segment. They generate answers by querying the model 10 times repeatedly. When disagreements arise, they invoke the model an additional 9–11 times (depending on the question type) to produce supplementary answers.
>
> [1] [Deep Video Discovery: Agentic Search with Tool Use for Long-form Video Understanding. NeurIPS. 2025.](https://openreview.net/forum?id=oQYq9L1NVT)
>
> [2] [Vgent: Graph-based Retrieval-Reasoning-Augmented Generation For Long Video Understanding. NeurIPS. 2025.](https://openreview.net/forum?id=5xPvWat3IX)
>
> [3] [From Trial to Triumph: Advancing Long Video Understanding via Visual Context Sample Scaling and Self-reward Alignment. ICCV, 2025.](https://openaccess.thecvf.com/content/ICCV2025/html/Suo_From_Trial_to_Triumph_Advancing_Long_Video_Understanding_via_Visual_ICCV_2025_paper.html)
>
> [4] [Adavideorag: Omni-contextual adaptive retrieval-augmented efficient long video understanding. NeurIPS. 2025.](https://openreview.net/forum?id=FDAI0PY9Qp)
>
> >The method was not applied to any state-of-the-art proprietary models
>
> Currently, the few training-free methods that have evaluated performance on closed-source large models typically reduce both the baseline frame count and the number of frames sampled in their experiments—usually to fewer than 20 frames. In our table, we directly report the best performance achieved by closed-source large models when using the maximum allowable number of frames. In the current mainstream literature, evaluation on three open-source models is generally considered to provide reasonably comprehensive validation. Following your suggestion, we are now seeking third-party APIs for closed-source large models that support inputting a larger number of frames.

---

> ### Author Response · Authors · 2025-11-21
> **Response to Reviewer ZkXQ (Part 2/2)**
>
> > Could you provide the experimental results of the proposed method on open-ended tasks?
>
> Thank you for your suggestion. Our designed ACR module is currently best suited for single-select multiple-choice questions. Most existing benchmarks for MLLM-based long-video understanding are primarily in the form of multiple-choice questions. In future work, we plan to explore benchmarks that support open-ended answers. For open-ended questions, we would obtain two candidate answers: Answer 1 and Answer 2. In the ACR prompt, we would phrase the instruction as follows:
>
> “< Question > + <Consider whether the answer could possibly be Answer 1 or Answer 2?>”
>
> > Experimental results on SVCFS (TAS-FS + FGMI-FS) without ACR are removed from Table 2.
>
> Following your suggestion, we reviewed Table 2 and found that it is complete.The two sets of frames selected by TAS-FS and CgMI-FS are fed separately into the MLLM, yielding two initial results. These two results are independent of each other. Only for fewer than 15% of videos (on average) do the corresponding questions require processing through the ACR module to obtain a calibrated final answer. Without the ACR module, the results correspond exactly to the two middle rows in Table 2—namely, those of TAS-FS and CgMI-FS.
>
> > Detailed explanation on notation $s_i^{ind}, s_i^{con}$
>
> Thank you for pointing this out. we will address and refine it in our subsequent revisions.
>
> > Could the authors provide a qualitative example that visualizes this failure case?
>
> Example:
>
> Question: "What was the reason for the woman in the video crying?"
>
> A. She was unhappy with the man.
> B. She found the man in the video to be foolish.
> C. She felt foolish.
> D. Insufficient information to determine.
>
> Correct answer: **C**. If one were to select the top-64 most relevant frames solely based on textual similarity to the question (e.g., keywords like **“woman”** and **“crying”**), the model would likely choose **D**, due to insufficient contextual evidence. This happens because such a frame selection strategy **overly focuses on surface-level keyword matches** and **fails to capture the causal context** behind the woman’s emotional reaction. By partitioning the video’s frame indices into multiple segments and sampling frames from across these segments, **our approach ensures that the selected frames better reflect the narrative progression and underlying causes** of the woman’s behavior, thereby enabling more accurate reasoning.

---

### Official Review · Reviewer_D2HU · 2025-11-03

**Soundness:** 3
**Presentation:** 3
**Contribution:** 3
**Rating:** 4
**Confidence:** 5

**Summary:**

The paper proposes SeViCES, a training-free, model-agnostic framework to improve long video understanding in Video-LLMs by selecting better evidence frames and enforcing semantic–visual consensus. It has two main components: (1) The Semantic–Visual Consensus Frame Selection (SVCFS) module selects frames through a temporal-aware semantic branch that leverages LLM reasoning over
captions, and a cluster-guided visual branch that aligns embeddings with semantic scores via mutual information. The second part is the Answer Consensus Refinement (ACR) module, which further resolves inconsistencies between semantic- and visual-based predictions by fusing evidence and constraining the answer space. Experiments on VideoMME, MLVU, LongVideoBench, and LVBench with three Video-LLMs (Qwen2.5-VL, LLaVA-Video, InternVL2.5) show consistent accuracy gains.

**Strengths:**

1. Clear and timely problem focus (long video + Video-LLMs).
The paper tackles an important and under-served problem: long video QA where naive uniform sampling or brute-force token input is infeasible. The motivation (computational cost, diluted attention, loss of reasoning consistency) is well-argued and backed by recent prior work.

2. Conceptually neat “consensus-driven evidence selection” idea.
Combining semantic (caption + LLM reasoning) and visual (embedding + clustering + MI) signals for frame selection is a clean, intuitive idea.
The extra step of answer-level consensus (ACR) — treating disagreement between semantic- and visual-based predictions as a signal of evidence incompleteness — is novel and nicely bridges selection and reasoning.

3. Experiment results demonstrate the effectiveness of the proposed method.

**Weaknesses:**

1. My major concern is the computation cost and scalability of the proposed LLM-based scoring method. TAS-FS requires two LLM calls per frame (independent + temporal context), plus captioning via BLIP-2 for all frames. Thus the computational cost is significantly higher than other methods with simple LLM usage.

2. The computational cost is not experimented and discussed in this work. It is recommend to compare the runtime cost with some benchmarking methods.

**Questions:**

Please refer to the weakness

---

> ### Author Response · Authors · 2025-11-21
> **Response to Reviewer D2HU**
>
> Thank you very much for your recognition of the methodology in our paper. We have analyzed the computational cost as per your suggestion.
>
> We adaptively select the number of sampled frames based on video duration.
> BLIP-2 takes an average of 0.09 seconds per frame to generate captions and embeddings for videos. When scoring each frame individually, we employ a binning strategy to select multiple captions from frames that are sufficiently distant from one another, and input them together into the LLM. The LLM then outputs the scores for all selected frames in a single response formatted as a list. This approach avoids the need to invoke the LLM repeatedly for each individual frame.
>
> For fewer than 15% of videos on average, the corresponding questions require an additional inference step through constrained decoding in the Answer Consensus Refinement module.
>
> **Table 1: Evaluation Computational Cost on VideoMME Benchmark**
> | Method   | Total Frames Input to MLLM/LLM for Frame Selection | Number of MLLM Calls |
> |----------|----------------------------------------------------|----------------------|
> | DVD      | 5,544,000 (MLLM)                                         | 589,500              |
> | Vgent    | 2,772,000 (MLLM)                                         | 136,800              |
> | Suo et al.   | 1,144,800 (MLLM)                                         | 38,340               |
> | SeViCES (Ours) | 1,522,800 (LLM)                          | 5,700                |
>
> Treating MLLMs or LLMs as agents and invoking them multiple times within a method has become increasingly common in the field of long-form video understanding [4]. We compare the computational costs of several recent methods in **Table 1**.
>
> DVD [1] divides videos into 5-second segments and samples each segment at 2 FPS, feeding the frames into GPT-4.1 to generate captions and build a database. During the iterative reasoning loop, DVD employs OpenAI o3 for agent-based reasoning and planning, while using GPT-4.1 to perform visual question answering on a subset of frames.
>
> Vgent [2] first samples long videos at 1 FPS and segments them accordingly. MLLMs (e.g., Qwen2.5-VL) extract key entities, actions, and scenes from both the question and each sampled frame. They further decompose the original question into multiple sub-questions, feed each video segment into the MLLM to answer these sub-questions, and finally input the selected frames into the MLLM to generate the final answer.
>
> Suo et al.[3] uniformly splits long videos into 32 segments and performs bin sampling by randomly selecting one frame from each segment. They generate answers by querying the model 10 times repeatedly. When disagreements arise, they invoke the model an additional 9–11 times (depending on the question type) to produce supplementary answers.
>
> [1] Deep Video Discovery: Agentic Search with Tool Use for Long-form Video Understanding. NeurIPS. 2025.
>
> [2] Vgent: Graph-based Retrieval-Reasoning-Augmented Generation For Long Video Understanding. NeurIPS. 2025.
>
> [3] From Trial to Triumph: Advancing Long Video Understanding via Visual Context Sample Scaling and Self-reward Alignment. ICCV, 2025.
>
> [4] Adavideorag: Omni-contextual adaptive retrieval-augmented efficient long video understanding. NeurIPS. 2025.

---

### Comment · Area_Chair_VQQL · 2025-11-26
**Please check the author's response.**

Dear Reviewers,

This is a gentle reminder that the authors have posted their responses to your comments. Please take a moment to review their feedback and initiate a discussion with the authors if necessary.

Best regards,
AC

---

### Note · Authors · 2026-01-06

I have read and agree with the venue's withdrawal policy on behalf of myself and my co-authors.